# Left Ventricular Noncompaction and Congenital Heart Disease Increases the Risk of Congestive Heart Failure

**DOI:** 10.3390/jcm9030785

**Published:** 2020-03-13

**Authors:** Keiichi Hirono, Yukiko Hata, Nariaki Miyao, Mako Okabe, Shinya Takarada, Hideyuki Nakaoka, Keijiro Ibuki, Sayaka Ozawa, Naoki Yoshimura, Naoki Nishida, Fukiko Ichida

**Affiliations:** 1Department of Pediatrics, Graduate School of Medicine, University of Toyama, Toyama 930-0194, Japan; nijiiroongakutai@yahoo.co.jp (N.M.); macoron.coron@gmail.com (M.O.); toyamadaikensyu2013@gmail.com (S.T.); kmc.ash7-oe.iia03@maroon.plala.or.jp (H.N.); kibk9925@gmail.com (K.I.); sozawa34@med.u-toyama.ac.jp (S.O.); 2Legal Medicine, Graduate School of Medicine, University of Toyama, Toyama 930-0194, Japan; yhatalm@med.u-toyama.ac.jp (Y.H.); nishida0717@yahoo.co.jp (N.N.); 3First Department of Surgery, Graduate School of Medicine, University of Toyama, Toyama 930-0194, Japan; ynaoki@med.u-toyama.ac.jp; 4Department of Pediatrics, International University of Health and Welfare, Tokyo 107-0052, Japan; fkichida@iuhw.ac.jp

**Keywords:** left ventricular noncompaction, congenital heart disease, congestive heart failure, non-ischemic cardiomyopathy, genetics

## Abstract

Background: Left ventricular noncompaction (LVNC) is a hereditary cardiomyopathy that is associated with high morbidity and mortality rates. Recently, LVNC was classified into several phenotypes including congenital heart disease (CHD). However, although LVNC and CHD are frequently observed, the role and clinical significance of genetics in these cardiomyopathies has not been fully evaluated. Therefore, we aimed to evaluate the impact on the perioperative outcomes of children with concomitant LVNC and CHD using next-generation sequencing (NGS). Methods: From May 2000 to August 2018, 53 Japanese probands with LVNC (25 males and 28 females) were enrolled and we screened 182 cardiomyopathy-associated genes in these patients using NGS. Results: The age at diagnosis of the enrolled patients ranged from 0 to 14 years (median: 0.3 months). A total of 23 patients (43.4%) were diagnosed with heart failure, 14 with heart murmur (26.4%), and 6 with cyanosis (11.3%). During the observation period, 31 patients (58.5%) experienced heart failure and 13 (24.5%) developed arrhythmias such as ventricular tachycardia, supraventricular tachycardia, and atrioventricular block. Moreover, 29 patients (54.7%) had ventricular septal defects (VSDs), 17 (32.1%) had atrial septal defects, 10 had patent ductus arteriosus (PDA), and 7 (13.2%) had Ebstein’s anomaly and double outlet right ventricle. Among the included patients, 30 underwent surgery, 19 underwent biventricular repair, and 2 underwent pulmonary artery banding, bilateral pulmonary artery banding, and PDA ligation. Overall, 30 genetic variants were identified in 28 patients with LVNC and CHD. Eight variants were detected in *MYH7* and two in *TPM1.* Echocardiography showed lower ejection fractions and more thickened trabeculations in the left ventricle in patients with LVNC and CHD than in age-matched patients with VSDs. During follow-up, 4 patients died and the condition of 8 worsened postoperatively. The multivariable proportional hazards model showed that heart failure, LV ejection fraction of < 24%, LV end-diastolic diameter z-score of > 8.56, and noncompacted-to-compacted ratio of the left ventricular apex of > 8.33 at the last visit were risk factors for survival. Conclusions: LVNC and CHD are frequently associated with genetic abnormalities. Knowledge of the association between CHD and LVNC is important for the awareness of clinical implications during the preoperative and postoperative periods to identify the populations who are at an increased risk of additional morbidity.

## 1. Introduction

Left ventricular noncompaction (LVNC) is the most recently classified cardiomyopathy. First described in 1990, it is characterized by a pattern of thickened trabeculations and deep intertrabecular fossa communicating with the left ventricular (LV) cavity [1]. LVNC has a wide spectrum, ranging from asymptomatic to severe congestive heart failure (CHF) with concomitant risks of arrhythmia, systemic thromboembolization, and sudden cardiac death. Although its diagnosis has primarily focused on the identification of trabeculation, other features are important to classify the specific subtypes of LVNC [2]. The specific phenotype of LVNC has the risk of adverse clinical outcomes in children and may occasionally be seen in association with congenital heart disease (CHD) [3].

The etiology of LVNC in patients with CHD has been unknown [4]. Previously, LVNC occurs most frequently in patients with Ebstein’s anomaly, following with septal defects, LV outflow tract obstructive lesions, hypoplastic left heart syndrome, and other right heart lesions [3,4,5]. However, the natural history of patients with LVNC and CHD has not been fully elucidated. In addition, it is considered difficult to estimate prognosis and to identify the surgical indications in children with LVNC because they have highly variable clinical presentations. Therefore, the aim of our study was to evaluate the impact on the perioperative outcomes of patients with concomitant LVNC and CHD.

## 2. Methods

### 2.1. Subjects

From May 2000 to August 2018, 53 Japanese probands with LVNC and CHD were referred to our institution for genetic testing from several institutions in Japan. Patients aged < 18 years diagnosed as LVNC at the participating institutions were eligible for this study. Patients with secondary etiologies of cardiomyopathy (e.g., endocrine, rheumatic, pulmonary, and immunologic diseases; systemic hypertension; and cardiotoxic exposures), those with a pacemaker because of rhythm disturbance, and those with non-follow-up records were excluded. Clinical data were retrospectively retrieved from the patients’ medical records according to the following time course: initial visit, preoperation, postoperation, and last visit. Cardiac death, LV assist device implantation, heart transplant (HT), and implantable cardioverter-defibrillator (ICD) shock were classified as major adverse cardiac events (MACEs).

Age matched patients with ventricular septal defects (VSDs) were selected from the Toyama University Hospital Database for comparison. All these patients underwent a surgery during the same period to endorse similarity of medical management.

Informed consent was gained from all patients or their guardians according to the institutional guidelines. The study protocol conformed to the ethical guidelines of the 1975 Declaration of Helsinki as reflected in the *a priori* approval of the Research Ethics Committee of the University of Toyama, Japan.

### 2.2. Endpoint Assessment 

Primary outcome was the time to the combined endpoint of MACEs, whereas secondary outcomes were arrhythmia, thromboembolic events, echocardiographic parameters, and genetic status.

### 2.3. Electrocardiogram Collection

All electrocardiograms were assessed independently by two well-trained investigators (K.H. and N.M.) who were blinded to the clinical data. The two investigators judged more than 95% consistency. The final judgment was made by a third experienced investigator (electrophysiologist) in cases of disagreement. The criteria for J wave and fragmented QRS were based on the description by Antzelevitch, Yan, and Das [6,7]. 

### 2.4. Echocardiographic Data Collection

Echocardiography (two-dimensional, color Doppler, and M-mode) was performed to assess cardiac structure, ventricular size and function (fractional shortening and ejection fraction (EF)), and valvular regurgitation. The diagnosis of congestive heart failure (CHF) was defined by clinical findings of tachypnea, feeding difficulty, and cyanosis; cardiomegaly on chest radiography; and decreased LVEF on echocardiography. Cardiomegaly was defined as a cardiothoracic ratio of ≥ 0.55 (≥0.60 for patients aged less than 1 year) on chest radiography or LV end-diastolic diameter (LVDD) of ≥120% of the normal value on echocardiography.

Patients were diagnosed with LVNC based on the following criteria defined by Ichida et al.: (1) two-layered myocardium with a noncompacted-to-compacted (N/C) ratio of more than 2.0 at end diastole, (2) prominent endomyocardial trabeculations that are distributed in more than one LV wall segment, and (3) deep fossas filled with blood from the ventricular cavity on color Doppler imaging [8]. All echocardiographic records were analyzed by two reviewers (K.H. and S.O.).

The thickness of the LV wall and N/C ratio (N; the depth of trabecular recesses. C; compacted wall thickness) were measured according to previously reported methods to quantify the extent of the trabecular meshwork [9,10]. The thickness of the compacted layer in the LV posterior wall (LVPWC) and LVDD are represented as z-scores based on the body surface area [11].

N/C ratios of 5 LV wall segments at end diastole; the anterior, lateral, and posterior walls; and interventricular septum at the level of the papillary muscles in the short-axis view and the apex in the long-axis view were measured [12,13,14].

### 2.5. Variant Screening

After obtaining informed consent from the patients or their parents, DNA was isolated from a whole-blood or heart tissue sample from each patient. Next-generation sequencing (NGS) was performed with 182 cardiac disorder-related genes (Appendix A) using the Ion PGM System (Life Technologies, Carlsbad, CA, USA).

After all candidate pathogenic variants passed the selection criteria, to validate the result of NGS, Sanger sequencing was conducted. 

### 2.6. Data Analysis and Variant Classification

The gnomAD database and Human Genetic Variation Database (HGVD), which contain data from 1208 Japanese individuals, was used to determine the allelic frequency of all detected variants. All variants were filtered with a minor allele frequency (MAF) of ≥0.0005 among the gnomAD and HGVD population. To evaluate the pathogenicity of the variants, seven different *in silico* predictive algorithms were used (Appendix A). The pathogenicity of an identified variant was evaluated by the guidelines of the American College of Medical Genetics and Genomics [15]. 

### 2.7. Gene-Based Collapsing Test

A genic collapsing test was performed to confer risk genes of LVNC [16,17]. Fisher’s exact test was conducted for each gene in collapsing analysis with a nominal significance level of < 2.74 × 10^−4^ for the number of assessable genes according to Bonferroni’s correction.

### 2.8. Statistical Analysis

Continuous variables are presented as mean ± standard deviation, and medians and ranges as appropriate. Categorical variables are given as frequencies and percentages. Statistical analyses were conducted with the use of the JMP software (version 13; SAS Institute, Cary, NC, USA). Receiver operating characteristic curve analysis was performed to determine the optimum cutoff levels of the number of derivations obtained from electrocardiogram and echocardiographic data to predict MACEs. A *p*-value of <0.05 was considered statistically significant.

## 3. Results

### 3.1. Clinical and Cardiological Characteristics

Patient demographics and outcomes are shown in Table 1 and Figure 1. Overall, 53 patients (25 males and 28 females) were enrolled in this study. Their age at diagnosis ranged from 0 to 14 years (median: 0.3 months). The median follow-up period was 3.0 years (0.5–17.0 years). A total of 11 patients (20.8%) reported a family history of cardiomyopathy; 23 (43.4%) were diagnosed with CHF, 14 with heart murmur (26.4%), and 6 with cyanosis (11.3%). During the observation period, 31 patients (58.5%) experienced CHF and 13 (24.5%) developed arrhythmias such as ventricular tachycardia, supraventricular tachycardia, and atrioventricular block. No patient had a history of thrombosis. Other systemic malformations were observed in 10 patients (18.9%; Appendix A).

According to the type of CHD, 29 patients (54.7%) had VSDs, 17 (32.1%) had atrial septal defects, 10 had patent ductus arteriosus (PDA), and 7 (13.2%) had Ebstein’s anomaly and double outlet right ventricle.

A total of 30 patients underwent surgery at, on average, 11.2 months of age; 19 underwent biventricular repair (BVR); and 2 underwent pulmonary artery banding (PAB), bilateral PAB, and PDA ligation. Moreover, 13 patients (43.3%) were diagnosed with LVNC postoperatively (Figure 1).

Upon electrography, fragmented QRS was frequently observed in 16 patients (40.0%), followed by right bundle branch block (25.0%), T-wave abnormality (20.0%), Q wave (17.5%), J wave (17.5%), ST-segment depression (12.3%), and long QT syndrome (12.5%).

### 3.2. Genetic Analysis

After excluding common polymorphisms on the basis of variant frequencies reported in gnomAD and HGVD and *in silico* analysis predictions, we identified 30 rare exonic (25 missense and 2 frameshift indel) and 3 splice site variants classified as pathogenic or likely pathogenic (Table 2) in 28 patients with LVNC and CHD. Eight variants were detected in *MYH7*, two in *TPM1*, and one in *ACTC1*, *ANK2*, *COL4A1*, *DAAM1*, *DSG2*, *DSP*, *FGF16*, *FGFR2*, *HCN4*, *JUP*, *MYBPC3*, *MYH6*, *MYL2*, *PKP2*, *PRDM16*, *RYR2*, and *TAZ* each. Sarcomere gene variants accounted for 50.0%. All variants affected conserved amino acid residues. In addition, the genetic collapsing test showed that variants in *MYH7* (*p* = 2.104 × 10^−16^, ranked first) and *TPM1* (*p* = 1.356 × 10^−4^, ranked second) reached significance (adjusted alpha = *p* < 2.74 × 10^−4^), which strongly suggested that variants in both genes increase the risk of LVNC (Appendix A).

### 3.3. Cardiological Characteristics

Echocardiography showed lower EFs and more thickened trabeculations in LV in children with LVNC and CHD than in those with VSDs (Table 3 and Figure 2). The average LV posterior wall (LVPW) z-score at the last visit was significantly higher than that at the initial visit (*p* = 0.0482). The average z-score of LVPWC thickness at the last visit was significantly lower than that of the initial visit (*p* = 0.0061).

### 3.4. Characteristics of Patients with Adverse Events

Adverse events were observed in 4 patients, and 4 patients died (Table 4 and Figure 1). Although cardiac death, LV assist device implantation, HT, and ICD shock were classified as MACEs in this study, none of the patients underwent HT or LV assist device implantation or experienced ICD shock. Aside from 1 patient, 3 died at early infancy and never underwent surgery.

The condition of 8 patients worsened postoperatively (Table 5 and Figure 1), all patients had VSDs, 3 had variants in *MYH7*, 6 underwent BVR, and long-term medical therapy were required in all patients for myocardial dysfunction after their latest surgeries.

The analysis of the multivariable proportional hazards model revealed that CHF during follow-up, LVEF of < 24%, LVDD z-score of > 8.56, and N/C ratio of the LV apex of > 8.33 at the last visit were risk factors for survival without MACE occurrence (Table 6). Patients with LVNC and CHD had a worse prognosis than those with VSDs (Figure 3).

## 4. Discussion

LVNC is associated with CHD, ranging from PDA or atrial septal defects/VSDs to more severe diseases such as Ebstein’s anomaly [4]. Our study demonstrated three features: (1) pathogenic variants were identified in more than half of the patients; (2) patients with LVNC had lower EFs than those with VSDs throughout the study period; and (3) postoperative deterioration was observed in several patients.

A variety of genetic disorders are associated with LVNC, including Z-disk and sarcomere gene variants, mitochondrial disorders, and ion channel gene variants [18,19,20,21,22]. Thus, structural congenital malformations and impaired LV myocardial differentiation may be caused by genetic abnormalities. Additionally, for the development of LVNC in a patient with genetic variants, remarkable change of hemodynamic circulation in the fetus may be a cofactor [23]. In our results, variants in *MYH7* were most commonly identified and the variants significantly increase the risk of LVNC by rare variant collapsing analysis. The mechanisms by which variants in the *MYH7* gene induce LVNC remain unclear. Analyzing the positions of these variants against their amino acid location showed several hotspots wherein variants are more popular, which seemed to tend to be in key functional locations. We observed that all variants in *MYH7* associated with LVNC were found in the segment 1 domain. Moreover, enrichment of pathogenic variants was observed in the crucial functional domains of the ATP-binding domain [24,25,26]. It suggested that the majority of the identified variants affect the force output by affecting either regulation of the ATPase cycle, movement of the lever, or interaction between myosin and actin. Remarkably, the location of variants in *MYH7* in LVNC was different from that in hypertrophic cardiomyopathy (HCM) patients. Hotspots of HCM were mostly located in the surface spanning the converter domain and the myosin mesa; the flat surface of the myosin catalytic domain [27]. It is important to distinguish variant types and assess them in light of well-known disease mechanisms. Therefore, to understand the pathophysiology and development of LVNC, it is critically notable for the patients with LVNC and CHD to characterize genetic variants and phenotypic abnormalities.

Children with LVNC and CHD have a higher incidence of CHF than patients with VSDs. Additionally, our results showed that the condition of 8 of the 30 patients (26.7%) worsened postoperatively, whether palliative or radical. The underlying pathophysiologic mechanisms of CHF remain unclarified. Systolic dysfunction in LVNC is believed to be due to subendocardial hypoperfusion [1,28]. It is also believed that diastolic dysfunction occurs by a restrictive filling pattern and abnormal relaxation because of the presence of LV hypertrabeculation [29]. These speculations are based on the evidence that the noncompact layer has typically low perfusion, which has been demonstrated on multiple modalities [30]. Functionally, LV torsion is more common in patients with LVNC [31]. LV twist is generated by the movement of two orthogonally oriented muscular bands of a helical myocardial structure concomitant with a clockwise rotation of the base and counterclockwise rotation of the apex in LV [32]. Van Dalen et al. used two-dimensional speckle tracking echocardiography and demonstrated that LV basal and apical rotation are in the same direction, resulting in a lack of LV twist in patients with LVNC [33]. Bellavia et al. reported that in adults, the value of LV rotation/torsion excessively decreased in patients with LVNC, whereas normal EF were retained when compared with those in controls [31]. Nawaytow et al. reported that almost half of the children with LVNC exhibit reverse apical rotation, resulting in decreased LV torsion and untwist rate, which are associated with the degree of LVNC [34]. These previous studies might support that the deterioration of LV function occurs during the perioperative period, although LV systolic function was preserved preoperatively because of its unique structure. 

In our study, 13 patients were diagnosed with LVNC postoperatively. The existence of additional triggers such as dynamic hemodynamic changes during the perioperative period were suggested. Indeed, the etiology of LVNC remains unknown. One possibility is that primary abnormality in early myocardial morphogenesis may cause LVNC. Another possibility is that prenatal or postnatal additional triggers such as pressure overload on the LV may cause LVNC. LVNC in the setting of CHD may be one of the models where both hemodynamic and genetic factors interact with each other, resulting in abnormal LV differentiation Thus, additional stress to the myocardium may trigger the worsening of systolic function in patients with LVNC and CHD because LVNC is more frequently associated with systolic dysfunction than that of CHD without LVNC. 

There were no predictors of postoperative CHF in this study. Preoperative preserved systolic function did not predict the outcome of patients with LVNC and CHD. In fact, not all patients with LVNC had systolic dysfunction during the preoperative evaluation. Most patients with LVNC and CHD had mildly depressed systolic function preoperatively. These facts may complicate the establishment of medical treatment during the operative period and optimal timing of surgery. 

In our study, patients with LVNC and CHD were observed to have a higher frequency of arrhythmias (24.5%). Recently, it was demonstrated in pediatric patients that LVNC with associated CHD confers additional risk [35]. Although there were no data relationships between mortality and the prevalence of arrhythmias in patients with LVNC and CHD, our data suggest that more attention should be paid to the occurrence of arrhythmia and CHF because of the higher prevalence of these symptoms.

### Limitations

The number of patients in our study was small. We were not able to track patients for a long period of time, particularly those who were referred from external facilities, because this was a retrospective study. This study included data from over approximately 15 years. During this period, the development of disease-modifying treatments was improved, which may have altered study outcomes. The small number of patients with genetic variants also has a limitation regarding the significance of the association with variants and prognosis. To clarify the effect of variants on clinical manifestations and prognosis, further analyses of larger numbers of patients are required. Additionally, functional analyses are also required to clarify the significance of the identified variants which contribute to the etiology of LVNC.

## 5. Conclusions

To the best of our knowledge, this is the first large cohort study to reveal the etiology and genetic background of LVNC and CHD. Recognition of the association between LVNC and CHD is crucial considering the increased risk of CHF as demonstrated in our results. Moreover, our data suggest that concomitant LVNC with CHD is a surgical risk factor, so additional perioperative planning such as CHF treatment may be beneficial if identified preoperatively. Therefore, elucidation of genotype-phenotype correlation in patients with LVNC and CHD may be important to understand the pathophysiology and development of LVNC in patients with CHD. Further studies will continue to determine long-term and genotype–phenotype correlations.

## Figures and Tables

**Figure 1 jcm-09-00785-f001:**
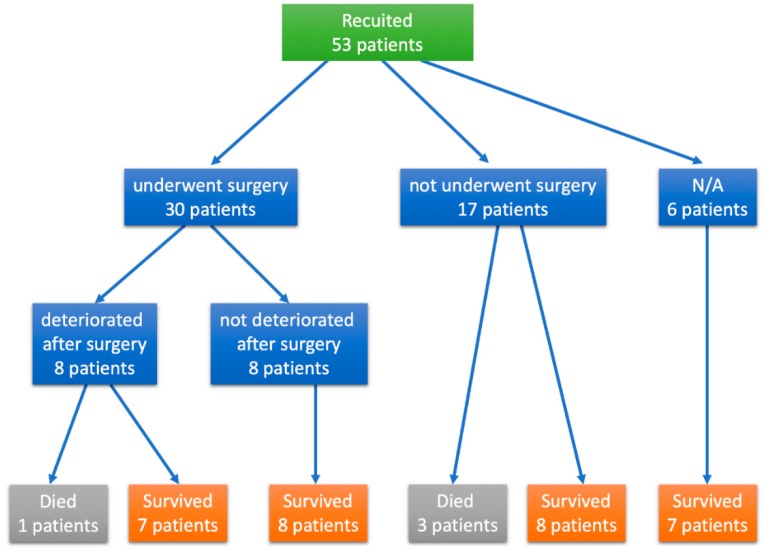
Flowchart of included and excluded patients. Thirty patients underwent surgery and the condition of 8 worsened postoperatively. Adverse events were noted in 4 patients.

**Figure 2 jcm-09-00785-f002:**
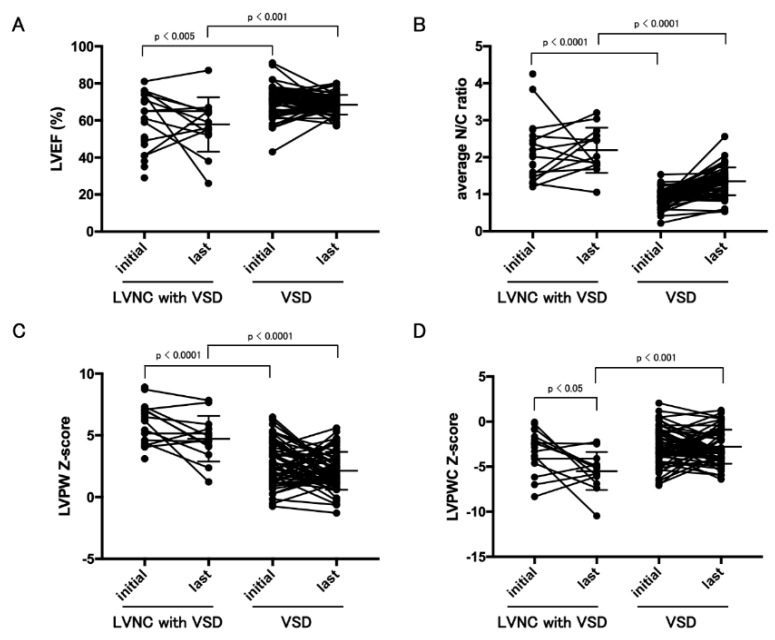
Longitudinal data of echocardiographic data between LVNC with CHD and VSD groups. LVEF (**A**), average N/C ratio (**B**), LVPW z-score (**C**), and LVPWC z-score (**D**) between the initial and last visits. LVNC; left ventricular noncompaction, VSD; ventricular septal defect, LVEF; left ventricular ejection fraction, N/C; ratio of noncompacted/compacted layer, LVPW; left ventricular posterior wall, LVPWC; compacted layer of left ventricular posterior wall.

**Figure 3 jcm-09-00785-f003:**
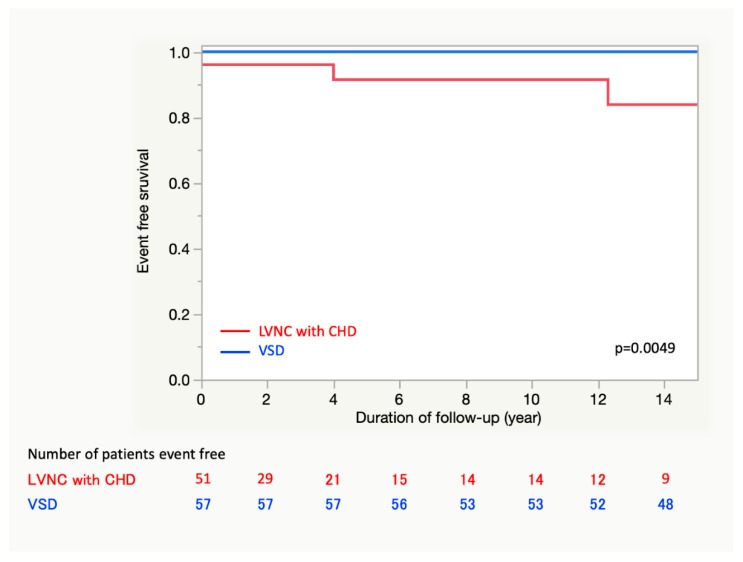
Event-free survival to the endpoint of major adverse cardiac events in LVNC with CHD and VSD groups. LVNC; left ventricular noncompaction, VSD; ventricular septal defect.

**Table 1 jcm-09-00785-t001:** Comparison of physical findings of patients with LVNC and VSD.

	LVNC with CHD (*n* = 53)	LVNC with VSD (*n* = 25)	VSD (*n* = 57)	*p* Value
Demographic data				
Male	25 (47.1%)	15 (60%)	29 (50.9%)	0.4724
Age at diagnosis (m, median)	0.27 (0–168)	0.4 (0–60)	0.10 (0–3)	0.3659
Duration of follow up (year)	3.0 (0.5–17)	3.0 (1.0–17)	3.9 (0.5–12.2)	0.6803
Family history of CM	11 (20.8%)			
Symptoms at diagnosis				
Heart failure	23 (43.4%)	11 (44.0%)	4 (7.0%)	0.0002
Heart murmur	14 (26.4%)	8 (32.0%)	50 (87.7%)	<0.0001
Cyanosis	6 (11.3%)	1 (4.0%)	0 (0%)	0.3086
Fetal screening	5 (9.4%)	3 (12.0%)	1 (1.8%)	0.0822
Neonatal screening	4 (7.5%)	1 (4.0%)	2 (3.5%)	0.6696
Arrhythmia	1 (1.9%)	1 (4.0%)	0 (0%)	0.3049
Current clinical presentation				
Heart failure	31 (58.5%)			
Embolism	0 (0%)			
Arrhythmia	13 (24.5%)			
VT	5 (9.4%)			
SVT	2 (3.8%)			
CAVB	4 (7.5%)			
AFL	2 (3.8%)			
Extracardiac abnormalities	10 (18.9%)			
HF requiring hospitalization	44 (83.0%)			
Age at surgery	11.15 ± 14.78			
Death	4 (7.5%)	1 (4.0%)	0 (0%)	0.2963
Type of CHD				
VSD	29 (54.7%)	25 (100%)	57 (100%)	<0.0001
ASD	17 (32.1%)			
PDA	10 (18.9%)			
Ebstein’s disease	7 (13.2%)			
DORV	7 (13.2%)			
CoA	4 (7.5%)			
PS	4 (7.5%)			
AS	2 (3.8%)			
hypo RV	2 (3.8%)			
absent PV	1 (1.9%)			
IAA	1 (1.9%)			
Heterotaxy syndrome	1 (1.9%)			
MA	1 (1.9%)			
PA	1 (1.9%)			
TA	1 (1.9%)			
TGA	1 (1.9%)			
TOF	1 (1.9%)			
TS	1 (1.9%)			
Type of surgery				
BVR	19 (35.8%)			
PAB	2 (3.8%)			
bil PAB	2 (3.8%)			
PDA ligation	2 (3.8%)			
PTPV	2 (3.8%)			
PTAV	1 (1.9%)			
PA debanding + PA plasty	1 (1.9%)			
Electrocardiography				
ST-segment depression	3 (7.5%)			
T-wave abnormality	8 (20.0%)			
Pathologic Q-wave	7 (17.5%)			
LBBB	3 (7.5%)			
RBBB	10 (25.0%)			
fragmented QRS	16 (40.0%)			
J wave	7 (17.5%)			
LQT	5 (12.5%)			
WPW syndrome	1 (2.5%)			

LVNC-CHD; left ventricular noncompaction with congenital heart disease, VSD; ventricular septal defect, CM; cardiomyopathy, VT; ventricular tachycardia, SVT; supra ventricular tachycardia, CAVB; complete AV block, AFL; atrial flatter, HF; heart failure, ASD; atrial septal defect, PDA; patent ductus arteiosus, Ebstein; Ebstein’s anomaly, DORV; double outlet of right ventricle, CoA; coarctation of aorta, PS; pulmonary valve stenosis, AS; aortic valve stenosis, hypo RV; hypoplastic right ventricle, PV; pulmonary valve, IAA; interruption of aortic arch, MA; mitaral vavle atresia, PA; pulmonary valve atresia, TA; tricuspid valve atresia, TGA; transposition of the great arteries, TOF; tetralogy of Fallot, TS; tricuspid valve stenosis, BVR; biventricular repair, PAB; pulmonary artery banding, bil PAB; bilateral pulmonary artery banding, PTPV; percutaneous transvenous pulmonary valvuloplasty, PTAV; percutaneous transvenous artery valvuloplasty, LBBB; left bundle branch block, RBBB; right bundle branch block, LQT; long QT. Continuous variables between the group of LVNC with VSD and the group of VSD were compared using the unpaired t-test, non-parametric Mann–Whitney test, or one-way analysis of variance, and categorical variables were compared using χ^2^ statistics or Fisher’s exact test, as appropriate.

**Table 2 jcm-09-00785-t002:** Variants identified in patients with LVNC.

Patient Number	Gene	Protein	Coding	dbSNP	gnomAD	gnomAD (EA)	HGVD	Fathmm	Sift	Polyphen2	GVGD	Mutation Taster	Provean	CADD	Clin Var
1	*LDB3*	p.Thr282Met	c.845C > T	n/a	0.00001991	0.0001088	n/a	n/a	0	1	C0	1	−0.28	−4.91	uncertain significance
2	*DSP*	p.Arg907Cys	c.2719C > T	n/a	0.00003185	0	n/a	−3.43	0	0.999	C0	1	1.96	−3.58	uncertain significance
2	*PKP2*	p.Thr50fs	c.148_151delACAG	rs397517067	n/a	n/a	n/a	−4.54	n/a	n/a	n/a	n/a	n/a	n/a	n/a
6	*JUP*	p.Arg233Pro	c.698G > C	n/a	n/a	n/a	n/a	−4.39	0.03	0.999	C0	1	1.03	−4.36	n/a
9	*MYL2*	p.Glu88Lys	c.262G > A	rs753032598	n/a	n/a	n/a	−0.23	0.017	0.996	C15	1	−1.15	−3.62	n/a
11	*MYH7*	p.Arg712His	c.2135G > A	rs199473346	n/a	n/a	n/a	−5.47	0	0.988	C25	1	−4.54	−4.35	n/a
16	*MYH7*	p.Met362Arg	c.1085T > G	rs199473346	n/a	n/a	n/a	−5.47	0	0.001	0	1	−3.64	−5.15	n/a
18	*MYH7*	p.Met362Arg	c.1085T > G	rs199473346	n/a	n/a	n/a	−5.47	0	0.001	0	1	−3.64	−5.15	n/a
20	*MYBPC3*	p.Arg891Trp	c.2671C > T	rs200229074	0.0000271	0.0001996	n/a	−1.83	0.001	1	C65	1	0.37	−6.52	uncertain significance
21	*TPM1*	p.Arg238Gln	c.713G > A	n/a	n/a	n/a	n/a	−3.32	0.001	0.999	C35	1	−6.36	−3.22	n/a
23	*MYH7*	−	c.896-1C > T	n/a	n/a	n/a	n/a	−4.49	n/a	n/a	n/a	n/a	n/a	n/a	n/a
24	*MYH7*	p.Phe230Ser	c.689T > C	n/a	n/a	n/a	n/a	5.9	0	0.984	0	1	−4.96	−6.07	n/a
25	*DAAM1*	p.Ala187fs	c.557_558insA	n/a	n/a	n/a	n/a	n/a	n/a	n/a	n/a	n/a	−	−	n/a
27	*MYH7*	p.Leu693Arg	c.2078T > G	rs749051278	n/a	n/a	n/a	1.96	0	0.997	C65	1	−4.85	−5.29	n/a
28	*MYH7*	p.Leu620Pro	c.1859T > C	n/a	0.000003977	0	n/a	n/a	7	0.969	C65	1	−4.06	−6.25	n/a
30	*MYH7*	p.Arg23Trp	c.67C > T	rs749297714	0.00002475	0	n/a	3.31	0.002	0.997	0	1	−2.08	−4.22	uncertain significance
31	*PRDM16*	p.Ser723fs	c.2168delC	n/a	0.00001071	0	n/a	−2.31	n/a	n/a	n/a	n/a	n/a	n/a	n/a
33	*TPM1*	p.Arg238Gln	c.713G > A	n/a	n/a	n/a	n/a	n/a	0.001	0.999	C35	1	−6.36	−3.22	n/a
34	*FGF16*	−	c.105 + 4AG > GT	n/a	n/a	n/a	n/a	n/a	n/a	n/a	n/a	n/a	n/a	n/a	n/a
34	*FGFR2*	−	c.939 + 40T > C	n/a	0.0000533	0.0004516	n/a	n/a	n/a	n/a	n/a	n/a	n/a	n/a	n/a
35	*COL4A1*	p.Pro108Ser	c.322C > T	rs769020772	n/a	n/a	n/a	n/a	0	0.999	C65	1	5.9	−5.28	n/a
36	*RYR2*	p.Leu4597Ser	c.13790T > C	n/a	n/a	n/a	n/a	n/a	0	0.999	C0	1	−5.09	−4.614	n/a
37	*ACTC1*	p.Arg212His	c.635G > A	rs121908411	n/a	n/a	n/a	−7.52	0	0.887	C25	1	−4.54	−3.896	uncertain significance
38	*TNNT2*	p.Lys298Thr	c.893A > C	rs121908411	n/a	n/a	n/a	−7.52	0	0.999	C0	1	2.19	−3.071	n/a
43	*ANK2*	p.Arg895Gln	c.2684G > A	rs146581757	0.00004773	0.0002719	0.04	−0.23	0.05	0.998	C0	1	−0.3	−3.277	n/a
45	*TAZ*	p.Gly216Arg	c.646G > A	n/a	n/a	n/a	n/a	1.03	0.05	1	C0	1	−3.5	−6.733	n/a
46	*MYH6*	p.Glu1713Lys	c.5137G > A	rs121908441	0.00003197	0.00005013	n/a	−2.4	0	1	C55	1	−1.84	−2.522	uncertain significance
47	*DSG2*	p.Tyr235His	c.703T > C	rs199472921	n/a	n/a	n/a	−4.93	0	1	C65	1	−0.41	−4.678	n/a
48	*COL4A1*	p.Gln462Arg	c.1385A > G	rs147445322	0.00001991	0.0002718	0.04	−6.37	0.04	0.491	C0	1	−3.19	−1.633	n/a
50	*HCN4*	p.Asp432His	c.1294G > C	rs147445322	n/a	n/a	n/a	−2.61	0.12	1	C0	1	−4.92	−5.923	n/a

**Table 3 jcm-09-00785-t003:** Comparison of physical findings of patients with LVNC and VSD between the initial and last visit.

	LVNC with VSD (*n* = 21)			VSD (*n* = 57)		
	Initial	Last	*p* Value	Initial	Last	*p* Value
Age	5.43 ± 13.24	56.25 ± 76.97	0.2768	19.05 ± 28.26	33.39 ± 28.32	<0.0001
Cardiac function at diagnosis						
LVEF (%)	59.00 ± 12.97	57.85 ± 14.73	0.7192	70.09 ± 8.29	68.47 ± 5.35	0.1363
LVDD Z-score	1.323 ± 3.16	1.14 ± 2.31	0.9576	2.02 ± 1.74	0.19 ± 1.00	<0.0001
LVPW Z-score	6.13 ± 1.57	4.72 ± 1.84	0.0482	2.73 ± 1.78	2.13 ± 1.53	0.0814
LVPWC Z-score	−3.13 ± 2.22	−5.48 ± 2.11	0.0061	−2.67 ± 2.07	−2.78 ± 1.89	0.5709
N/C ratio						
Anterior wall	−3.13 ± 2.22	−5.48 ± 2.11	0.5370	0.50 ± 0.31	0.38 ± 0.18	0.0632
Septal wall	−3.13 ± 2.22	−5.48 ± 2.11	0.1723	0.41 ± 0.20	0.29 ± 0.11	0.0015
Lateral wall	−3.13 ± 2.22	−5.48 ± 2.11	0.9385	0.73 ± 0.44	0.72 ± 0.37	0.8735
Posterior wall	3.21 ± 1.45	3.81 ± 1.50	0.2028	1.30 ± 0.50	1.14 ± 0.49	0.0846
Apex	3.45 ± 1.72	3.97 ± 1.62	0.1913	1.52 ± 0.88	4.22 ± 1.62	<0.0001
Mean 5 segments	2.13 ± 0.84	2.19 ± 0.61	0.4897	0.89 ± 0.24	1.35 ± 0.38	<0.0001

LVNC-CHD; left ventricular noncompaction with congenital heart disease, VSD; ventricular septal defect, LVEF; left ventricular ejection fraction, LVDD; left ventricular diastolic dimension, LVPW; left ventricular posterior wall, LVPWC; compacted layer of left ventricular posterior wall, N/C; ratio of noncompacted/compacted layer.

**Table 4 jcm-09-00785-t004:** Summary of death cases.

ID	Sex	Age at Diagnosis	Symptoms at Diagnosis	CHD	Variants	FH of CM	Extracardiac Abnormalities	Current Clinical Presentation	Age at Death	Cause of Death
13	F	1 d	bradycardia	VSD	-	yes	no	CHF	1d	shock
43	M	0 d	CHF	DORV, TGA, VSD	*ANK2*	no	no	CHF	12y	CHF
45	M	0 d	CHF	ASD, PDA	*TAZ*	no	no	CHF	2m	shock
52	M	fetus	CHF	Ebstein	-	no	no	CHF	0d	shock

CHD; congenital heart disease, FH; family history, CM; cardiomyopathy, VSD; ventricular septal defect, DORV; double outlet of right ventricle, Ebstein; Ebstein’s anomaly, ASD; atrial septal disease, TVD; tricuspid valve dysplasia. BVR; biventricular repair, CHF; congestive heart failure.

**Table 5 jcm-09-00785-t005:** Summary of deteriorated cases postoperatively.

ID	Sex	Age at Diagnosis	Symptoms at Diagnosis	CHD	Variants	FH of CM	Extracardiac Abnormalities	Type of Surgery	Age at Surgery	Current Clinical Presentation	Outcome
5	M	1 d	cyanosis	DORV, VSD, IAA	-	no	no	BVR	5	CHF	alive
16	F	fetus	cyanosis	Ebstein, VSD, CoA	*MYH7*	yes	no	BVR	1	CHF	alive
17	F	1 y	CHF	VSD	-	no	no	BVR	16	CHF	alive
18	F	4 d	heart murmur	Ebstein, VSD, CoA	*MYH7*	yes	no	BVR	0	CHF	alive
23	F	fetus	CHF	DORV, VSD, IAA	*MYH7*	yes	no	bilPAB	0	CHF	alive
43	M	0 d	cyanosis	TGA, DORV, VSD	*ANK2*	no	no	BVR	17	CHF	death
47	F	12 d	heart murmur	ASD, VSD, PDA	*DSG2*	no	chromosome 12 abnomality	PA debanding + PA plasty	34	CHF	alive
53	M	1 m	CHF	VSD	-	no	no	BVR	1	no	alive

CHD; congenital heart disease, FH; family history, CM; cardiomyopathy, DORV; double outlet of right ventricle, VSD; ventricular septal defect, IAA; interruption of aortic arch, Ebstein; Ebstein’s anomaly, CoA; coarctation of aorta, TGA; transposition of the great arteries, ASD; atrial septal disease, PDA; patent ductus arteiosus, BVR; biventricular repair, PAB; pulmonary artery banding, bil PAB; bilateral pulmonary artery banding.

**Table 6 jcm-09-00785-t006:** Univariate analysis of risk factors for death in the patients with LVNC.

	Univariable Survival Analysis	
Variable	Odds Ratio (95% CI)	*p* Value
Male	3.54 (0.42–74.51)	0.2536
Family history	4.33 (0.47–40.33)	0.1809
Heart failure at diagnosis	4.39 × 10^7^ (0.62–)	0.1049
Heart failure	8.06 × 10^6^ (1.13–)	0.0404
Extracardiac abnormalities	4.75 (0.51–44.78)	0.1586
Gene variants	0.85 (0.96–7.52)	0.8726
Double variants	2.87 × 10^−6^ (–5.89 × 10^−121^)	0.5676
UCG parameters at first visit		
LVEF < 50%	0.89 (0.098–8.13)	0.9160
UCG parameters at last visit		
LVEF < 24%	1.84 × 10^15^ (0.0051–10.55)	0.0051
LVDD Z score > 8.56	1.84 × 10^15^ (0.0054–9.95)	0.0054
N/C ratio of apex > 8.33	1.84 × 10^15^ (0.0051–10.55)	0.0051

CI, confidence interval, UCG; cardiac ultrasound, LVEF; left ventricular ejection fraction, LVDD; left ventricular diastolic dimension, N/C; ratio of noncompacted/compacted layer.

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
