# Peer review of "Left Ventricular Noncompaction and Congenital Heart Disease Increases the Risk of Congestive Heart Failure"

_jcm, 2020, doi:10.3390/jcm9030785_

Round 1

Reviewer 1 Report

The combination of congenital heart failure due to congenital heart disease and non compaction left ventricle is well known. A variety of congenital heart diseases are described. Significant arrhythmia, death and complications depend of left ventricular function, end-diastolic diameter and for sure non compaction cardiomyopathy. Although the number of patients included in this paper is small, significance is positive.   

Author Response

Thank you for your kindly constructive comments and your consideration of the revised version.

Sincerely,

Reviewer 2 Report

The authors report their clinical, imaging, and genetic analysis of 53 Japanese children with left ventricular noncompaction (LVNC) associated with congenital heart disease and compared the features (imaging and clinical), major adverse cardiac events (MACE), and outcomes. The patient cohort included 87% with septal defects (VSD 55%, ASDD 32%), 10% PDA, and 13% had Ebstein anomaly, most typically associated with DORV. Approximately 59% of patients developed heart failure. Genetic analysis identified 30 rare exonic variants that were classified as pathogenic or likely pathogenic, with MyH7 being most common. Echocardiography identified lower ejection fractions and more prominent trabeculation burden in the LVNC-CHD group compared to a VSD-only cohort. Four children died and four had MACE. Eight patients had clinical decline post-operatively, all with VSDs (+/- other defects), with 6 patients undergoing biventricular repair. Outcomes compared with the VSD-only cohort was worse.

The manuscript is well written and well discribed. There are several points that should be addressed.

  1. The VSD-only comparison group would be expeccted to have a better outcome and fewer symptoms compared to those with VSD associated with other cardiac defects (such as Ebstein, DORV, etc). Did the authors verify outcomes compared to children with similar defects and no evidence of LVNC and who underwent similar surgeries?
  2. The spread of clinical phenotypes seems to be without some of the other forms of CHD such as HLHS than many cohorts. Is there a reason for that? 

Author Response

We thank the reviewer for his/her careful reading. We have addressed the reviewer’s questions as follows.

Comment 1:

The VSD-only comparison group would be expected to have a better outcome and fewer symptoms compared to those with VSD associated with other cardiac defects (such as Ebstein, DORV, etc). Did the authors verify outcomes compared to children with similar defects and no evidence of LVNC and who underwent similar surgeries?

>> Eighty patients with atrial septal defects (ASDs), 24 patients with tetralogy of Fallot (TOF), and 10 patients with atrioventricular septal defects (AVSD) were matched for age and selected from the Toyama University Hospital Database for comparison. All these patients underwent a surgical procedure during the same era to ensure similarity of management practices and personnel over time. As a result, there was no death in these patients during the study period.

Comment 2:

The spread of clinical phenotypes seems to be without some of the other forms of CHD such as HLHS than many cohorts. Is there a reason for that?

>> We enrolled 53 Japanese probands with LVNC and CHD who were referred to our institution for genetic testing from several hospitals in Japan. Therefore, the difference of population of the patients with CHD from other previous studies might come from the difference of race or ethnicity. Another possibility was that the present study was conducted over approximately 15 years. During the study period, the detection rate of LVNC and CHD might change because LVNC and CHD was a fairly uncommon 15 years ago.

Thank you for your kindly constructive comments and your consideration of the revised version.

Sincerely,